# Impact of Sexual Dimorphism on Therapy Response in Patients with Metabolic Dysfunction-Associated Steatotic Liver Disease: From Conventional and Nutritional Approaches to Emerging Therapies

**DOI:** 10.3390/nu17030477

**Published:** 2025-01-28

**Authors:** Eleonora Dileo, Francesca Saba, Mirko Parasiliti-Caprino, Chiara Rosso, Elisabetta Bugianesi

**Affiliations:** Department of Medical Sciences, University of Turin, 10126 Turin, Italy; eleonora.dileo@unito.it (E.D.); francesca.saba@unito.it (F.S.); mirko.parasiliticaprino@unito.it (M.P.-C.)

**Keywords:** metabolic dysfunction-associated steatotic liver disease, metabolic dysfunction-associated steatohepatitis, sexual dimorphism, steroids, sex hormones, therapy

## Abstract

Metabolic dysfunction-associated steatotic liver disease (MASLD) represents a spectrum of liver disease ranging from hepatic fat accumulation to steatohepatitis (metabolic dysfunction-associated steatohepatitis, MASH), fibrosis, cirrhosis, and potentially hepatocellular carcinoma in the absence of excessive alcohol consumption. MASLD is characterized by substantial inter-individual variability in terms of severity and rate of progression, with a prevalence that is generally higher in men than in women. Steroids metabolism is characterized by sexual dimorphism and may have an impact on liver disease progression; indeed, several therapeutic strategies targeting hormone receptors are under phase 2/3 development. Despite the fact that the importance of sexual dimorphism in the setting of MASLD is well recognized, the underlying molecular mechanisms that can potentially drive the disease toward progression are not clear. The aim of this review is to delve into the crosstalk between sexual dimorphism and steroid hormone perturbation under nutritional and pharmacological intervention.

## 1. Introduction

Metabolic dysfunction-associated steatotic liver disease (MASLD) is a liver disease strongly associated with metabolic comorbidities such as obesity, type 2 diabetes mellitus (T2DM), arterial hypertension, and dyslipidemia. Liver damage encompasses a wide and complex spectrum, ranging from simple steatosis to the progressive metabolic dysfunction-associated steatohepatitis (MASH), characterized by ballooning degeneration and lobular inflammation with or without fibrosis [1]. In this context, hepatic fibrosis is the most important negative prognostic factor, driving MASLD toward cirrhosis, which, in turn, increases the risk of hepatocellular carcinoma (HCC) [1]. MASLD prevalence ranges from 25% to 30% in the adult European population [2] and is projected to be the principal etiology for liver transplantation within the decade [2,3]. Despite the high prevalence, only a subset of patients with MASLD progress to advanced liver disease over a period of years, ultimately experiencing major morbidity and mortality [4,5]. In fact, MASLD is characterized by an inter-patient variability in severity and rate of progression, as well as bi-directional changes in grade of MASH and stage of liver fibrosis, as disease severity ‘waxes and wanes’ over time [6]. The determinants of an individual’s risk of MASH and advanced fibrosis/cirrhosis or HCC are still not clear.

The liver function/physiology is characterized by sexual dimorphism, and the pathogenic mechanisms involved in the onset and progression of liver disease differ, at least in part, according to sex and sex-related factors [7], as seen in Figure 1. This observation has been further discussed in a recent large-scale analysis of hepatic transcriptomic profiles that stressed the sexually dimorphic nature of MASLD and its association with the progression of liver disease, suggesting that sex should be always considered as one of the main non-modifiable risk factors for patients’ stratification [8]. Susceptibility to MASLD is also influenced by a complex interplay between environmental and genetic factors [9]. In the past fifteen years, genome-wide association studies (GWAS) led to the identification of several single nucleotide polymorphisms (SNPs) that are linked to hepatic steatosis and progressive forms of the disease. The majority of the SNPs affect genes encoding for proteins involved in lipid metabolism. Specifically, the risk variants in *PNPLA3*, *TM6SF2*, *MBOAT7*, *GCKR* and the protective variant in *HSD17B13* genes are mostly related to the lipid handling in the hepatocytes and are associated with MASLD onset and progression [10,11,12]. The genetic variant rs72613567:TA in the gene *HSD17B13*, associated with a reduced risk of MASH and fibrosis, encodes an uncharacterized member of the hydroxysteroid 17-beta dehydrogenase family. The 17beta-hydroxysteroid dehydrogenases are important enzymes in steroid metabolism, particularly in the regulation of the biological effects of steroid hormones, but these enzymes are also involved in the regulation of fatty acid metabolic pathways, affecting several bioactive lipid species (e.g., leukotriene B4), that have been implicated in lipid-mediated inflammation [13]. Although the exact mechanism by which this SNP protects against MASLD is still unclear, anti-*HSD17B13* therapies are ongoing in the setting of clinical trials [14,15].

Unhealthy lifestyle behaviors contribute to weight gain in both sexes, with a predominantly subcutaneous localization in women and visceral localization in men. Overweightness/obesity increase the cardiometabolic risk through the development of type 2 diabetes, arterial hypertension, dyslipidemia, and hepatic steatosis. Mediterranean and the low-carbohydrate diets are highly recommended to reduce weight and improve metabolic derangement and cardiovascular risk. However, the physiological decrease in estrogens levels later in life predisposes women to a metabolic perturbation that may increase the risk of cardiometabolic events again. Abbreviations: CVD, cardiovascular disease; HDL, high density lipoprotein cholesterol; IR, insulin resistance; LCD, low-carbohydrate diet; MASLD, metabolic-dysfunction associated steatotic liver disease; MeD, Mediterranean diet; SAT, subcutaneous adipose tissue; T2DM, type 2 diabetes mellitus; TG, triglycerides; VAT, visceral adipose tissue.

Among the modifiable risk factors, sedentary lifestyles and changes in dietary patterns have contributed to the dramatic increase in MASLD prevalence, together with the rapidly progressing epidemics of both adult and childhood obesity [16]. In this regard, lifestyle intervention is considered the first-line approach to induce weight loss and to improve liver injury by histology or non-invasive biomarkers, as suggested by the last published joint Clinical Practice Guidelines on the management of MASLD [17]. In this review, we aimed to summarize the available data on the role of sexual dimorphism, in terms of hormonal perturbation, during the onset and progression of MASLD and its impact on conventional and emerging therapies for MASH.

## 2. The Importance of Sexual Dimorphism in MASLD: The Relevance of Steroids Metabolism

MASLD is a sexually dimorphic disease with a prevalence that is generally higher in men compared to women. Consistently, the prevalence of MASH, advanced hepatic fibrosis, as well as the occurrence of liver-related complications, is 2/4-fold higher in men than women [18]. However, the molecular mechanisms underlying sexual dimorphism in MASLD remains an important unmet question. Glucocorticoids and sex hormones (estrogens and androgens) are primarily responsible for differences among sexes also through the modulation of both glucose and lipid metabolism, but they can exert detrimental effects when their values fall outside the physiological range [19]. Alterations in sex hormones levels in both sexes are associated with lipid metabolism derangement, insulin resistance, T2DM, and hepatic steatosis. Consistently, hormone replacement/deprivation therapies may reverse MASLD [20]. Looking at the metabolic differences between men and women, we observe that premenopausal women display a better metabolic profile compared to men and postmenopausal women, pointing out on the protective effect of estrogens. This effect is mediated by decreasing the rate of lipolysis, the flux of free fatty acids (from the adipose tissue to other organs, including the liver), de novo lipogenesis, and by increasing fatty acids oxidation, thus reducing lipid storage and hepatic steatosis [21]. In women, estrogens prevent metabolic syndrome and the development of cardiovascular diseases (CVD), while low levels of estrogens influence the onset of MASLD [22,23]. In particular, the decrease in estrogens levels after menopause is associated with a high risk of developing MASH and severe hepatic fibrosis regardless of the most representative confounders (i.e., age, weight, visceral adiposity) and several studies reported a similar prevalence of MASLD and MASH in men and postmenopausal women [24,25,26]. Furthermore, postmenopausal women show a higher prevalence of advanced fibrosis (F ≥ 2) compared to fertile women, while conflicting data on the impact of estrogens in the progression of liver disease are reported in the latter group [27,28]. Sexually dimorphic role of androgens in human metabolic diseases is an emerging topic, with female androgen excess and male androgen deficiency sharing an overlapping adverse metabolic phenotype, including abdominal overweight/obesity, glucose metabolism impairment, and MASLD [29]. The beneficial effects of androgens are well described in both animal and human studies in which their physiological levels are able to protect against MASLD preventing or attenuating the features of metabolic syndrome [20]. Conversely, some androgenic species such as dehydroepiandrosterone (DHEA) and dehydroepiandrosterone sulfate (DHEAS) are associated with insulin resistance and with the most important histological features of MASLD [29,30,31]. High levels of androgens in women exert a detrimental effect on metabolic pathways and can be associated with other diseases. For example, hyperandrogenism is a hallmark in women with polycystic ovary syndrome (PCOS). The prevalence of MASLD in women with PCOS ranges from 15% to 55%, while the prevalence of PCOS in women with MASLD reaches 77%, suggesting that androgen excess in women likely play a role in the onset of MASLD [32,33]. However, the pathogenetic role of androgens in the onset and worsening of MASLD remains to be fully explored. According to the hypogonadal–obesity–adipokine hypothesis, the excess of adipose tissue promotes the conversion of testosterone to estradiol via aromatase activity. Estradiol inhibits the hormone kisspeptin which, in turn, impairs testosterone production. This mechanism is exacerbated by leptin that is released in large amounts from the inflamed and hypertrophic adipose tissue, impacting the gonadal axis, thus impairing testosterone production. Low testosterone levels promote adipocytes differentiation and inflammation, thus enhancing insulin resistance [34,35].

Therapeutic approaches in the setting of MASLD are focused on the improvement of metabolic abnormalities by targeting different metabolic pathways (including lipids, hormones, etc.). In this context, sex is not always considered for patients’ risk stratification in clinical trials; therefore, in the following paragraphs, we will summarize the most important studies describing the sexual dimorphism in response to conventional and emerging therapies.

## 3. Impact of Sexual Dimorphisms on Conventional and Emerging Therapeutic Strategies

### 3.1. Diet-Based Treatment

Sedentary lifestyle, low fiber diet, high intake of simple sugars, saturated fats, and ultra-processed foods have dramatically contributed to the epidemic of obesity, T2DM, and steatotic liver disease over the years [36]. Several studies demonstrated the benefit of a healthy lifestyle in reducing hepatic steatosis [37,38,39]. However, the optimal dietary regimen for preventing and managing MASLD in both men and women remains to be defined and, to date, sex-specific nutritional interventions are scanty. From a nutritional perspective, dietary patterns may modulate the risk of metabolic diseases in a sex-specific manner since many organs and tissues are characterized by sexual dimorphism [40,41]. Moreover, distribution of body fat composition has important differences according to gender, with a prevalent subcutaneous localization in pre-menopausal women and with a visceral localization in men [42]. Specifically, visceral adiposity in men is associated with an increased risk of metabolic diseases such as dyslipidemia, T2DM, hepatic steatosis, and CVD. Conversely, CVD risk is lower in pre-menopausal women compared with men [42]. Sex hormones have an impact on fat distribution and cardiometabolic risk as demonstrated in studies on female-to-male transsexual patients, who lose gluteal-femoral fat and gain visceral adipose tissue after the appropriate hormone replacement therapy [43,44]. In addition, as compared to subcutaneous fat, visceral adipocytes are characterized by a high expression of glucocorticoid receptors that, along with a high activation of the hypothalamic–pituitary–adrenal axis, is responsible of the preferential fat storage in visceral depots associated with insulin resistance and with an increased risk of CVD [45,46]. In women, lipolytic activity in the adipose tissue is reduced and post-load fatty acids are stored in the subcutaneous fat as a reservoir. Conversely, visceral fat, which predominates in men, is more metabolically active and more susceptible to lipolysis. This different energy distribution/storage is also due to the different physiological role during life, as men spend most of their energy on muscular activity, whereas women store energy to maintain reproductive capacity when energy reserves are limited [47,48].

Westerterp-Plantenga et al. explored the effect of diet switch (high protein to carbohydrate) in both sexes, showing that the high-protein diet was associated with greater energy expenditure and fatty acid oxidation in men and with high satiety in women [49]. A study on 782 subjects with overweight/obesity (506 women and 277 men) who underwent a low-calorie diet for six months showed that pre-menopausal women were less responsive to the diet compared with men in terms of cardiometabolic risk parameters such as triglycerides and total cholesterol levels [50]. Similarly, other studies have shown the long-term beneficial effects of the Mediterranean diet on cardiovascular health in men, who maintained a better metabolic profile over time compared with women [51,52,53]. This sexual dimorphism is supported by the decrease in estrogens levels over time that in post-menopausal women leads to an impairment in lipid metabolism that is typical during the physiological transition toward menopause. Vitale et al. showed that the regulation of glucose homeostasis in 156 adults at high risk of developing T2DM, randomized to receive a high vs. low glycemic index diet, was linked to sex-differences in body composition (fat distribution) and energy metabolism. Specifically, they observed that women were more sensitive to metabolic response due to the protective effects exerted by estrogens [49]. As with other dietary regimens, the ketogenic diet seems to be more effective in men compared with women and, even in this case, sex difference is due to the effect exerted by estrogens [54,55]. A study performed on 42 subjects with obesity who underwent a very low-energy ketogenic nutritional intervention for 45 days displayed no differences among sexes since all the participants significantly reduced weight and improved their inflammatory status. However, after sex and age stratification, men showed a better response in terms of body weight loss compared with fertile women and, among women, those with menopause showed the worst response to the intervention [54]. A summary of the studies described in the paragraph is reported in Table 1.

From a basic research point of view, the study of the interplay between sexual dimorphism and diet intervention in murine models is biased by the fact that the majority of the studies are limited to the male sex because of the variability of estrogens levels during estrous cycles in females [67,68,69]. However, studies performed on both sexes revealed conflicting results compared with those observed in humans pointing out the importance of age stratification. For example, a study aimed at exploring sex and age differences in mice fed with a ketogenic diet showed that male mice were more predisposed to develop glucose intolerance and insulin resistance regardless of age, while the reverse happened in female mice who were characterized by an increase in CVD risk but were protected against the risk of developing diabetes [70]. There are several preclinical studies on the effects of a Western diet on both sexes, and the overall results suggest that young male mice are more susceptible to diet-induced obesity when fed with a high fat diet. Conversely, young female mice show a greater capacity to use fat as a source of fuel, increasing energy expenditure, and displaying a better immune response when exposed to a fat diet [71,72,73]. Data on aged animals are less clear, suggesting a worse response to a high fat diet over time for both sexes. Specifically, old females gain weight and worsen glucose profile more easily compared with males, and this is in part due to the changes in sex hormones [74,75,76]. All the evidence suggests the importance of sex and age stratification to understand differences in diet response in preclinical studies. Overall, a regular healthy and varied diet with an adequate nutrient distribution seems to be the best option for improving metabolism in subjects with steatotic liver disease and metabolic comorbidities, probably contributing to the physiological range of hormone levels, thus mitigating their detrimental effects on metabolic pathways.

### 3.2. Antihyperglycemic Treatment

Sexual dimorphism plays an important role in the pathophysiology of MASLD through insulin resistance and adiposity [77,78]. Generally, men receive a diagnosis of T2DM earlier in life compared to women. Conversely, the prevalence of T2DM in women increases after menopause. The pathophysiologic explanation of these sex specific differences is mainly the loss of estrogens later in life and its impact on glucose metabolism [79]. Knock-out (KO) mice for the estrogen receptor alpha (ERα) (LERKO model) show hyperglycemia during fasting as a consequence of the increased gluconeogenic activity [75]. In the aromatase KO model, sexual dimorphism is more evident, since only male mice show glucose intolerance and insulin resistance, which are restored after estrogens replacement [80,81,82]. This was also demonstrated in T2DM rat models using estradiol (E2) treatment, which increases insulin levels through the G protein-coupled estrogen receptor (GPER) [83]. Similarly, androgens are able to modulate glucose metabolism. High testosterone levels are associated with an increased risk of T2DM in women compared with men [83]. In addition, androgens improve insulin sensitivity in both sexes, enhancing glucose utilization in the skeletal muscle [84]. A sexual dimorphism has also been described for the androgens receptor (AR) in pancreatic b-cells. Specifically, the testosterone-AR complex in males is able to enhance insulin secretion by potentiating the insulinotropic effect of the glucagon-like peptide 1 (GLP-1), while in females, the excess of androgens led to insulin hypersecretion, thus promoting mitochondrial dysfunction and oxidative injury [85,86].

Among the anti-hyperglycemic drugs, GLP-1 receptor agonists (RAs) are a class of glucose-lowering drugs approved for the treatment of T2DM [87]. Specifically, in patients with T2DM, GLP-1 RAs prevent the occurrence of clinical events over time such as cardiovascular outcomes, diabetes complication, and all-cause mortality by improving glycemic control and by reducing body weight and insulin resistance [87,88]. The main therapeutic goal in the management of T2DM is the reduction in glycated hemoglobin (HbA1c) levels, since HbA1c levels lower than 7% are associated with a lower risk of complications over time [89]. Several studies show that the reduction in Hb1Ac during exenatide or dulaglutide treatment is similar in men and women and depends on baseline Hb1Ac levels [57,58,90]. Conversely, when administered in combination with metformin, exenatide exerts a stronger effect in women compared with men, as shown by the reduction in Hb1Ac levels after treatment (from 8.8% to 6.8% and from 8.9% to 7.5%, respectively), suggesting better glycemic control in women compared with men [57]. However, when stratified by age, women using liraglutide treatments show better glycemic control in the subgroup of those younger than 65 years compared to the counterparts, underlining the importance of age-related estrogens reduction after menopause [59]. The latter consideration is sustained by a study demonstrating that lower baseline levels of Hb1Ac are associated with a better glycemic control in men but not in women, in which previous treatment of metformin seems to prevent treatment failure with GLP-1 RA [91]. GLP-1 RA induces weight loss and has also been approved for the treatment of obesity [60]. Several studies show that body weight reduction during treatment with exenatide is more pronounced in women compared to men, and similar results are also reported the same results after dulaglutide and liraglutide treatment [58,59,60,61,90,91]. In a recent study, Pelusi et al. showed that in men with low testosterone levels, treatment with semaglutide is safe and may restore androgens levels mainly through the reduction in body weight [92]. The molecular basis of this sexual dimorphism can be explained at least in part by the increased drug exposure observed in women, which is about 30% higher compared to men of comparable weight, suggesting that female sex may be an independent predictor of weight loss [93].

### 3.3. Lipid Lowering Treatment

Dyslipidemia is a common metabolic dysfunction in subjects with MASLD with a prevalence ranging from 60% to 70% [94]. This atherogenic profile (characterized by high levels of low-density lipoproteins cholesterol (LDL-C), low levels of high-density lipoprotein cholesterol (HDL-C), and high levels of triglycerides) is associated with an increased risk of developing CVD over time regardless of sex [95]. However, pre-menopausal women develop CVD about 10 years later compared to men, highlighting once again the protective role that estrogens exert on lipid metabolism [96]. In addition, women are more responsive to statin treatment compared with men [97]. From a mechanistic point of view, sexual dimorphism may be in part explained by the complex interplay between sex hormones and cardiovascular risk. For example, single nucleotide polymorphisms (SNPs) in the gene *CYP19A1*, encoding for the aromatase enzyme that is involved in the final step of estrogen synthesis in both men and postmenopausal women, are associated with hypertension, apolipoprotein B levels, and insulin resistance [98]. In addition, high estradiol levels in fertile women are associated with an increased expression of LDL receptors and this interaction contributes to the improvement in lipid levels during lipid-lowering treatment [99]. Estrogen signaling pathways may also increase the hepatic reverse transport of cholesterol in women, thus promoting its removal from peripheral tissues and reducing cardiovascular risk [98]. Finally, estrogens may recognize highly conserved genomic sequences, namely estrogen response elements (EREs), in the promoter of several genes by regulating their expression, thus modulating metabolic pathways (as described later in the paragraph on nucleotide-based therapies and *PNPLA3* gene), as show in Figure 2 [100].

Statins therapy in MASLD is indicated to prevent CVD, even if their efficacy for treating MASH remains to be proved in large randomized control trials with histological endpoints [101]. The efficacy of statins in improving the plasma lipid profile of women with dyslipidemia is lower compared to men and this can be at least in part explained by the fact that drugs such as atorvastatin and simvastatin are predominantly metabolized by CYP3A4, which is more expressed in women compared to men [62]. In addition, statins improve inflammatory status modulating the expression of several pro-inflammatory cytokines such as interleukin-6 (IL-6), tumor necrosis factor-alpha (TNF-α), and C reactive protein and, also in this case, women are more responsive than men [97]. A sexual dimorphism has also been described for the occurrence of adverse events. Women experienced myalgia and toxic skeletal muscle injury more frequently compared to men, while the reverse has been observed about liver function impairment [63,102]. This sexual dimorphism can be explained by the different pharmacokinetics that, in turn, depend on the distribution of body fat that affects drug availability [103].

Proprotein convertase subtilisin/kexin type 9 (PCSK9) is a key regulator of cholesterol metabolism that increases LDL-C levels through the degradation of the LDL receptors. High PCSK9 concentrations are associated with MASLD and with an increased risk of CVD. PCSK9 inhibitors are able to block the degradation of LDL receptors, thus increasing LDL-C uptake from the circulation [103,104,105]. Recent studies have observed different PCSK9 levels among sexes, suggesting a related sexual dimorphism. In addition, PCSK9 concentrations differed according to age and to the presence of metabolic abnormalities. However, clinical trials did not show evidence about the existence of sexual dimorphism in response to PCSK9 inhibitors, confirming the same strong effect in reducing LDL-C in both sexes [64,65]. Further studies are needed to better understand the impact of sexual dimorphism on lipid lowering response.

### 3.4. Nucleotide-Based Therapies

It is well known that genetic background contributes to the onset and progression of MASLD [106]. Genome-wide association studies have identified several genes involved in the regulation of lipid metabolism (i.e., *PNPLA3*, *TM6SF2*, *MBOAT7*, *GCKR*, *HSD17B13*) that contribute to the progression of liver damage [107]. Patatin-like phospholipase domain-containing 3 (*PNPLA3*) is a triacylglycerol lipase located on lipid droplets in both adipose tissue and liver, which is involved in the hydrolysis of triglycerides [108]. PNPLA3 protein expression is increased in subjects with obesity compared to those without, while the presence of the rs738409 SNP encodes for a loss of function variant that promote triglyceride retention in lipid droplets, favoring the development of hepatic steatosis and enhancing inflammation [109]. The association between the *PNPLA3* SNP and sexual dimorphism emerged over ten years ago in a meta-analysis with the aim to assess the strength of the effect of the *PNPLA3* rs738409 SNP on MASLD and the severity of liver disease across different populations. The analysis of 16 studies revealed the strong transversal association between genetic variants and MASLD [110]. Other studies showed a strong association among the rs738409 variant and increased ALT levels in men regardless of age and in women over 50 years. Considering the differences in estrogen levels in fertile and postmenopausal women, the authors hypothesized the existence of an interaction between the *PNPLA3* SNP and estrogens [110,111]. Recently, in a longitudinal study with a median follow-up of 7 years on a large European cohort of biopsied-proven MASLD patients, the authors found that women without obesity, older than 50 years, and carrying the rs738409 GG risk genotype, displayed a high risk of developing liver-related events over time compared to the other subgroups of patients [112]. This evidence points out the importance of risk stratification in the heterogeneous population of MASLD subjects and on the importance of *PNPLA3* genotyping as well as sex stratification for the identification of high-risk groups.

Over the last few years, oligonucleotide-based therapies such as small-interfering RNA (siRNA) and antisense oligonucleotide (ASO) have been developed for the treatment of MASLD by reducing the expression of the *PNPLA3* gene [113]. Preclinical studies showed that these new drugs are able to interfere with PNPLA3 synthesis by silencing the corresponding gene, thus reducing liver disease. For example, the hepatocyte-targeted N-acetylgalactosamine (GalNac)-conjugated ASO improves MASH and hepatic fibrosis in PNPLA3 rs738409 GG knock-in mice fed with a steatogenic diet; *PNPLA3* silencing through short-hairpin RNA improves liver steatosis in *PNPLA3* rs738409 GG knock-in mice fed a high fructose diet; the administration of lipid nanoparticles containing siRNA against human *PNPLA3* improves histological features of MASH including liver fibrosis, shown in Figure 3 [114,115,116,117]. In humans, the results of a phase 1 trial revealed that a single dose of GalNac-conjugated *PNPLA3* siRNA significantly reduced liver fat content [66,118], but no sexual dimorphism has been currently described in the treatment response. It is known that the *PNPLA3* gene is under nutritional control, through the liver X receptor and the steroid regulator element binding protein (SREBP)-1c pathway, which is involved in the regulation of fatty acid synthesis [119]. At the same time, estrogens can modulate lipogenic genes, such as those belonging to the SREBP family, promoting hepatic steatosis [120]. Sex hormones, like estrogens, also modulate lipogenic genes such as the SREBP family, changing lipid metabolism and then contributing to the onset of liver steatosis [116]. Recently, an interaction between estrogens and the *PNPLA3* genetic variant has been described, which seems to negatively affect the progression of MASLD [121]. Specifically, the authors found an increase in *PNPLA3* transcription in immortalized cell line and tissue-derived liver organoids treated with ERα agonists. After cells exposure to an excess of fatty acid, they observed a specific increase in the mutated protein with a contextual accumulation of intracellular lipids and collagen deposition. The removal of the *PNPLA3*-ERE by CRISPR/Cas9 led to a reduced response to ERα treatment, demonstrating the existence of a link between female sex hormones, *PNPLA3* SNP, and impaired lipid metabolism [121].

The mutant 148 Isoleucine to Methionine (I148M) protein variant in the patatin-like phospholipase domain-containing 3 (*PNPLA3*) gene results in a loss of function of the protein determining triglyceride retention in the hepatocytes, thus resulting in hepatic steatosis. The mutated protein is also resistant to ubiquitylation and accumulates in lipid droplets. Oligonucleotide-based therapies, such as small-interfering RNA and antisense oligonucleotide, have been developed in recent years for the treatment of MASLD by reducing the expression of the *PNPLA3* gene carrying the genetic variant. Abbreviations. ASO, antisense oligonucleotide; FFAs, free fatty acids; *PNPLA3*, patatin-like phospholipase containing domain 3; siRNA, small-interfering RNA; TG, triglycerides; VLDL, very low-density lipoproteins; RISC, RNA-induced silencing complex.

## 4. Conclusions

MASLD is a sexual dimorphic liver disease tightly associated with metabolic comorbidities, and a multidisciplinary therapeutic approach is required for managing these heterogeneous patients. Lifestyle interventions are highly recommended for all MASLD subjects regardless of sex, along with treatment of metabolic co-morbidities such as dyslipidemia and T2DM. However, pharmacological responses can be different among sexes. Sex hormones play an important role in the modulation of glucose and lipid metabolism, affecting also drug metabolism in the liver. For this reason, an appropriate consideration of sex and hormonal status could be useful to understand sexual dimorphism in MASLD risk and in response to pharmacological treatments. Finally, the study of the interplay between sexual dimorphism, hormones, and sexually dimorphic genes may help to identify novel potential therapeutic targets, leading to a more defined sex-specific personalized therapy.

## Figures and Tables

**Figure 1 nutrients-17-00477-f001:**
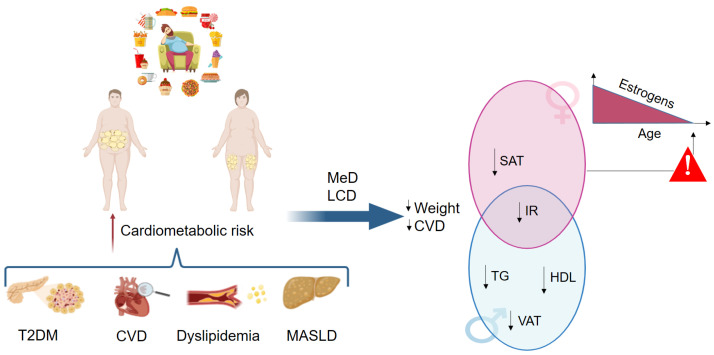
Impact of sexual dimorphism on the beneficial effects of diet.

**Figure 2 nutrients-17-00477-f002:**
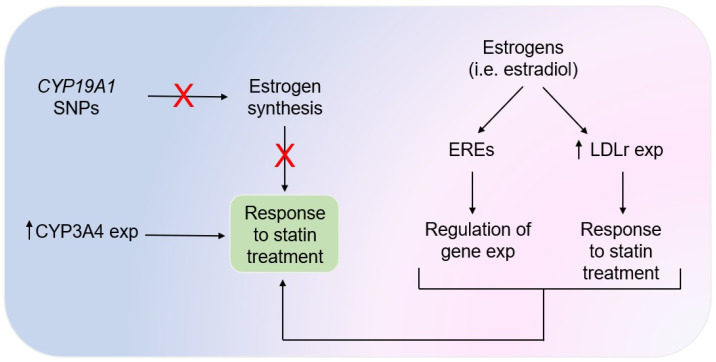
Interplay between sex hormones, lipid lowering drugs, and therapy response. Estrogens exert a protective effect on lipid metabolism in both men and women, but the latter group shows a better response to the statin treatment due to the favorable interaction between sexually dimorphic genes and drugs metabolism as well as the direct interaction between estrogens and sexually dimorphic genes. Abbreviations: *CYP19A1*, CYP19A1-cytochrome P450 family 19 subfamily A member 1 gene; *CYP3A4*, cytochrome P450 3A4 gene; EREs, estrogen response elements; exp, expression; LDLr, low-density lipoprotein receptor; SNPs, single nucleotide polymorphisms.

**Figure 3 nutrients-17-00477-f003:**
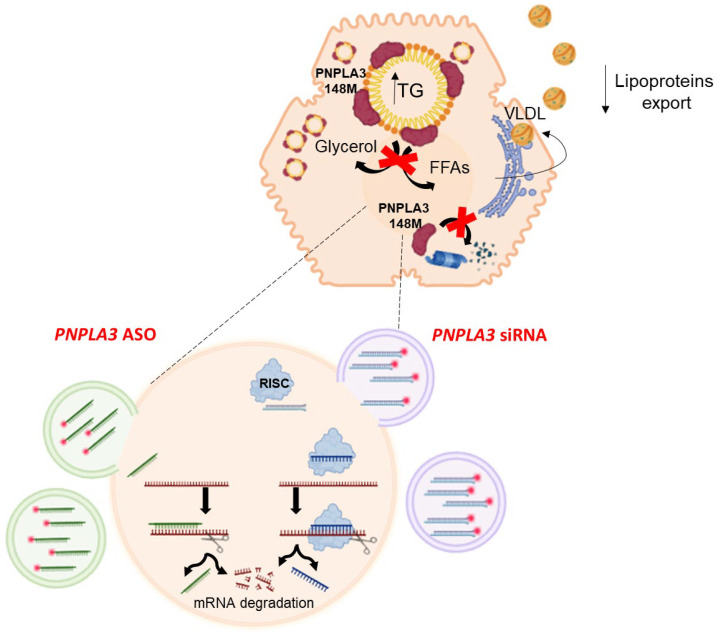
Antisense oligonucleotide and small-interfering RNA mechanisms of action.

**Table 1 nutrients-17-00477-t001:** Sexual dimorphism in response to conventional and emerging therapies. Abbreviations. F, females; M, males; na, not available.

Ref.	*n*, Sex: Age	Intervention and Overall Dosage	Conclusions
**Nutritional intervention**			
Trouwborst et al. [50]	*n* = 782 M: 276 (42.5 ± 6.0) F: 506 (41 ± 6.3)	6-month low-calorie diet	Women were less responsive to the diet compared with men in terms of cardiometabolic risk parameters
Bédard et al. [51]	*n* = 69 M: 37 (42.6 ± 7.3) F: 32 (41.2 ± 7.3)	4-week MeD	Men improved cardiovascular health through the reduction in postprandial insulin concentration
Leblanc et al. [52]	*n* = 123 M: 64 (41 ± 7.9) F: 59 (41.8 ± 6.7)	12-week MeD	Men showed a significant improvement of metabolic profile
Carruba et al. [53]	*n* = 115 Post-menopausal women	6-month MeD	Traditional Mediterranean diet significantly reduces endogenous estrogen in healthy postmenopausal women.
Vitale et al. [56]	*n* = 156 M: 74 (55.7 ± 10.7) F: 82 (55.1 ± 10.7)	12-week high- versus low-GI diet	Women, compared to men, showed a better metabolic improvement
D’Abbondanza et al. [54]	*n* = 70 M: 28 (20–62) F: 42 (17–67)	25-day very low-carbohydrate ketogenic diet	Men compared to pre-menopausal women, showed a better response in terms of body weight loss and improvement in MASLD
Muscogiuri et al. [55]	*n* = 42 M: 21 (37.7 ± 10.7) F: 21 (32.7 ± 8.7)	45-day very low-energy ketogenic diet	Men compared to women showed a better response in terms of weight loss and reduction in inflammation
**Antihyperglycemic treatment**			
Gallwitz et al. [57]	*n* = 3375 M: 1673 (56.8 ± 9.8) F: 1702 (55.9 ± 10.1)	Dulaglutide 1.5 mg and 0.75 mg for 12 months	Significant improvement in glycaemic control irrespective of gender
Onishi et al. [58]	*n* = 855 M: 649 (56.9 ± 10.1) F: 206 (58.7 ± 11.3)	Dulaglutide 0.75 mg for 26 weeks	Body weight reduction during treatment is more pronounced in women compared to men
Mirabelli et al. [59]	*n* = 40 (57.5 ± 6.6) M: 18 (na) F: 22 (na)	Liraglutide 1.2 mg or 1.8 mg for a minimum follow-up of 5 years	Prolonging treatment exerted a lasting benefit in women
Buysschaert et al. [60]	*n* = 184 (59 ± 11) M: 110 (na) F: 74 (na)	Exenatide 5 μg or 10 μg for 9 and 12 months	Body weight reduction during treatment with exenatide is more pronounced in women compared to men
Quan et al. [61]	*n* = 105 M: 51 (48.4 ± 9.8) F: 54 (49.1 ± 10.4)	Exenatide 5 μg + metformin 0.5 g for 4 weeks Exenatide 10 μg + metformin 0.5 g for 24 weeks	Combination therapy showed better results in women compared with men
**Lipid lowering therapy**			
Mombelli et al. [62]	*n* = 337 M: 171 (57.5 ± 9.2) F: 166 (59.4 ± 8.8)	Atorvastatin or rosuvastatin 10 mg/day Pravastatin and simvastatin 20 mg/day Fluvastatin 80 mg/day	Women compared with men showed a greater reduction in LDL-C
Smiderle et al. [63]	*n* = 495 M: 164 (59.9 ± 11.1) F: 331 (62.3 ± 10.7)	Simvastatin or atorvastatin for approximately 6 months	Adverse drug reactions were more frequent in women than in men
Sever et al. [64]	*n* = 27,564 M: 20,795 (62 ± 9) F: 6769 (64.1 ± 8.8)	Evolocumab or placebo for 2.2 years	Similar efficacy regardless of age in both men and women
Bittner et al. [65]	*n* = 18,924 M: 14,162 (mean age 57) F: 4762 (mean age 62)	Alirocumab 75 mg or placebo for a median follow-up of 2.8 years	Improvement of cardiovascular outcomes regardless of sex
**Nucleotide-based therapy**			
Fabbrini et al. [66]	Study 1: *n* = 55 (24–65) M: 24 (na) F: 31 (na) Study 2: *n* = 9 (38–61) M: 8 (na) F: 1 (na)	Study 1: Placebo or JNJ-0795 10/25/75/200/400 mg Study 2: Placebo or 75 mg JNJ-0795	A single dose of GalNac-conjugated *PNPLA3* siRNA reduced liver fat content, but no sexual dimorphism has been currently described concerning treatment response

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
