# Peer review of "Impact of Sexual Dimorphism on Therapy Response in Patients with Metabolic Dysfunction-Associated Steatotic Liver Disease: From Conventional and Nutritional Approaches to Emerging Therapies"

_nutrients, 2025, doi:10.3390/nu17030477_

Round 1
Reviewer 1 Report
Comments and Suggestions for Authors
The authors’s review focuses on the Impact of sexual dimorphism on therapy response in patients with metabolic dysfunction-associated steatotic liver disease.
NAFLD is a metabolic liver disease that encompasses a wide spectrum from simple steatosis to steatohepatitis (NASH) and fibrosis to cirrhosis and hepatocarcinoma. It has also been termed a “barometer of metabolic health” due to its metabolic roots. NAFLD is more prevalent in patients with pre-existing metabolic conditions than in the general population. Specifically, type 2 diabetes and NAFLD have a particularly close relationship.
The review highlights the protective effect of estrogens, aiming that fertile women are protected from NAFLD compared to men ,and NAFLD is more prevalent in post-menopausal women than pre-menopausal women and worsens after menopause However, women with polycystic ovary syndrome have a prevalence of NAFLD between 15% and 55%, especially when it is associated with hyperandrogenism 10.3748/wjg.v21.i39.11053; 10.3390/gastroent15040071 As hyperandrogenism is a hallmark of PCOS, androgens likely play a role in the development of NAFLD.
Therefore, this sexually dimorphic role of androgens in human metabolic disease is an emerging topic, with female androgen excess and male androgen deficiency sharing an overlapping adverse metabolic phenotype, including abdominal overweight/obesity, dysglycemia, insulin resistance, and NAFLD. 10.3390/ijms20112841
Low levels of testosterone and SHBG in men are independent predictors of the occurrence of metabolic syndrome. Furthermore, according to the hypogonadal-obesity-adipokine hypothesis, increased amounts of adipose tissue convert testosterone to estradiol via aromatase activity. Estradiol inhibits kisspeptin liberation and testosterone production. Moreover, adipose tissue produces leptin and proinflammatory cytokines that both have an effect on the gonadal axis, impairing testosterone production.
I suggest the authors discuss in their manuscript the relationship between diseases that affect the levels of estrogens/ androgens, both in women and in men, for a better approach to the involvement of sexual hormones in NAFLD pathogenesis, adding recommended references. Therefore, an appropriate consideration of sex and hormonal status could be useful to understand sexual differences in NAFLD risk and its response to pharmacological treatments.
Author Response
We would like to thank the Editors and the Reviewers for the positive and invaluable comments. We have revised the manuscript accordingly. All changes are reported in red. Below, our point-by-point response to the Reviewers’ comments:
Reviewer 1.
The authors’s review focuses on the Impact of sexual dimorphism on therapy response in patients with metabolic dysfunction-associated steatotic liver disease.
NAFLD is a metabolic liver disease that encompasses a wide spectrum from simple steatosis to steatohepatitis (NASH) and fibrosis to cirrhosis and hepatocarcinoma. It has also been termed a “barometer of metabolic health” due to its metabolic roots. NAFLD is more prevalent in patients with pre-existing metabolic conditions than in the general population. Specifically, type 2 diabetes and NAFLD have a particularly close relationship.
The review highlights the protective effect of estrogens, aiming that fertile women are protected from NAFLD compared to men ,and NAFLD is more prevalent in post-menopausal women than pre-menopausal women and worsens after menopause However, women with polycystic ovary syndrome have a prevalence of NAFLD between 15% and 55%, especially when it is associated with hyperandrogenism 10.3748/wjg.v21.i39.11053; 10.3390/gastroent15040071 As hyperandrogenism is a hallmark of PCOS, androgens likely play a role in the development of NAFLD.
Therefore, this sexually dimorphic role of androgens in human metabolic disease is an emerging topic, with female androgen excess and male androgen deficiency sharing an overlapping adverse metabolic phenotype, including abdominal overweight/obesity, dysglycemia, insulin resistance, and NAFLD. 10.3390/ijms20112841
Low levels of testosterone and SHBG in men are independent predictors of the occurrence of metabolic syndrome. Furthermore, according to the hypogonadal-obesity-adipokine hypothesis, increased amounts of adipose tissue convert testosterone to estradiol via aromatase activity. Estradiol inhibits kisspeptin liberation and testosterone production. Moreover, adipose tissue produces leptin and proinflammatory cytokines that both have an effect on the gonadal axis, impairing testosterone production.
I suggest the authors discuss in their manuscript the relationship between diseases that affect the levels of estrogens/ androgens, both in women and in men, for a better approach to the involvement of sexual hormones in NAFLD pathogenesis, adding recommended references. Therefore, an appropriate consideration of sex and hormonal status could be useful to understand sexual differences in NAFLD risk and its response to pharmacological treatments
R: We would like to thank the reviewer for his/her thoughtful comments that help us to significantly improve the quality of the manuscript. We have deeply explored the potential role of androgens in the pathogenesis of MASLD in paragraph 2 “The importance of sexual dimorphism in MASLD: the relevance of steroids metabolism” (line 102-106; line 114-131) and we have added all the significant references as suggested. In addition, we have included in the text a Table with the studies showing the impact of sexual dimorphism in response to diet and pharmacological treatments.
Reviewer 2 Report
Comments and Suggestions for Authors
In the present review, Dileo et al. highlight recent evidence demonstrating that, due to the protective effects of estrogens, the sex of the patient is relevant in the onset and progression of metabolic dysfunction-associated steatotic liver disease (MASLD). They also provide evidence demonstrating that men and women can have different responses to the same MASLD therapy. In women, being fertile or postmenopausal is also a relevant factor in the risk of developing MASLD and in its management. Therefore, the main concept of this study is that the sex of the patient (and the pre/post-menopausal status in women) is a relevant factor to understand and treat MASLD.
This topic is relevant to the MASLD field; however, its novelty is limited, since another review on this topic (cited in this manuscript) has been recently published: 10.1016/j.molmed.2024.05.013. This 2024 article is not in open-access, for which any similarity to the present manuscript could not be assessed. Nevertheless, sexual dimorphism is still a poorly explored factor in MASLD, which ensures some merit to the present work. Other related works are more focused (10.1016/j.tips.2024.05.004; 10.1186/s13293-024-00617-z) and less comprehensive than the present manuscript.
The main concept of this review is supported by the literature and the cited references are appropriate. The manuscript is very well-written and addresses the relevant aspects of its field. Figures are intuitive and properly described in their legends. There are no obvious missing aspects or inaccuracies other than those mentioned below:
1. The figures are helpful to understand the main concepts of the study; however, they were not referred to in the main text. Please mention each of the figures in the main text.
2. For clarity, in line 284, please include the full names of IL-6 and TNF-α.
Author Response
Reviewer 2.
In the present review, Dileo et al. highlight recent evidence demonstrating that, due to the protective effects of estrogens, the sex of the patient is relevant in the onset and progression of metabolic dysfunction-associated steatotic liver disease (MASLD). They also provide evidence demonstrating that men and women can have different responses to the same MASLD therapy. In women, being fertile or postmenopausal is also a relevant factor in the risk of developing MASLD and in its management. Therefore, the main concept of this study is that the sex of the patient (and the pre/post-menopausal status in women) is a relevant factor to understand and treat MASLD.
This topic is relevant to the MASLD field; however, its novelty is limited, since another review on this topic (cited in this manuscript) has been recently published: 10.1016/j.molmed.2024.05.013. This 2024 article is not in open-access, for which any similarity to the present manuscript could not be assessed. Nevertheless, sexual dimorphism is still a poorly explored factor in MASLD, which ensures some merit to the present work. Other related works are more focused (10.1016/j.tips.2024.05.004; 10.1186/s13293-024-00617-z) and less comprehensive than the present manuscript.
The main concept of this review is supported by the literature and the cited references are appropriate. The manuscript is very well-written and addresses the relevant aspects of its field. Figures are intuitive and properly described in their legends.
R: We would like to thank the reviewer for his/her positive comments.
There are no obvious missing aspects or inaccuracies other than those mentioned below:
- The figures are helpful to understand the main concepts of the study; however, they were not referred to in the main text. Please mention each of the figures in the main text.
R: We have referenced the figures throughout the manuscript.
- For clarity, in line 284, please include the full names of IL-6 and TNF-α.
R: We have added the full name of the cited acronyms.
Reviewer 3 Report
Comments and Suggestions for Authors
This manuscript summarized several potential impacts of sexual dimorphisms on therapies for patients with MASLD: 1. the diet-based treatment because male and female have different energy expenditure responses towards diets; 2. Antihyperglycemic treatment because higher testosterone/androgen is related to increased risk of T2DM; 3.lipid lowering treatment since women have a better response to the statin treatment because of estrogens. 4. nucleotide-based therapies since the authors hypothesized the existence of an interaction between PNPLA3 SNP and estrogens.
Sexual dimorphism of MASLD is not a new topic. There have been reviews published recently about sexual dimorphism of MASLD (PMID: 38890029), the molecular basis of sexual dimorphism for MASLD (PMID: 34417588) or therapeutic management of MASLD (PMID: 38193865). The current review is clear and relatively comprehensive, however, resmetirom should be discussed. Besides, there are a few problems with the current draft:
1) The beneficial effect of the diet exists in both male and female to different extent. Although men and women have different food choices, energy expenditure or nutrient metabolism rate and the progression of MAFLD. Personalized diet instruction is better for treatment. It should not be misled towards sexual dimorphism.
2) In the Phase 1 Trials of PNPLA3 siRNA in I148M Homozygous Patients with MAFLD, 44% of the participants were males. Based on the observation that estrogen receptor-α could interact with PNPLA3 p.I148M variant, it was an overstatement to say that PNPLA3 siRNA treatment has different responses towards male or female patients.
3) Sex-based dosage optimization should be considered.
Author Response
Reviewer 3.
This manuscript summarized several potential impacts of sexual dimorphisms on therapies for patients with MASLD: 1. the diet-based treatment because male and female have different energy expenditure responses towards diets; 2. Antihyperglycemic treatment because higher testosterone/androgen is related to increased risk of T2DM; 3.lipid lowering treatment since women have a better response to the statin treatment because of estrogens. 4. nucleotide-based therapies since the authors hypothesized the existence of an interaction between PNPLA3 SNP and estrogens. Sexual dimorphism of MASLD is not a new topic. There have been reviews published recently about sexual dimorphism of MASLD (PMID: 38890029), the molecular basis of sexual dimorphism for MASLD (PMID: 34417588) or therapeutic management of MASLD (PMID: 38193865). The current review is clear and relatively comprehensive, however, resmetirom should be discussed.
R: We would like to thank the reviewer for his/her positive comments.
Besides, there are a few problems with the current draft:
- The beneficial effect of the diet exists in both male and female to different extent. Although men and women have different food choices, energy expenditure or nutrient metabolism rate and the progression of MAFLD. Personalized diet instruction is better for treatment. It should not be misled towards sexual dimorphism.
R: We agree with the reviewer about the overall beneficial effect of diet. The most important limitation of these studies is that sex is not considered for the optimization of nutritional approach. We have added a Table in the text summerizing the main results of several studies focusing on sex differences in response to diet, mainly in terms of metabolic changes.
- In the Phase 1 Trials of PNPLA3 siRNA in I148M Homozygous Patients with MAFLD, 44% of the participants were males. Based on the observation that estrogen receptor-α could interact with PNPLA3 p.I148M variant, it was an overstatement to say that PNPLA3 siRNA treatment has different responses towards male or female patients.
R: We agree with the reviewer about his/her observation on the Phase 1 trial on PNPLA3 siRNA. In the text (lines 374-375), we stated that “…no sexual dimorphism has been currently described in the treatment response”.
- Sex-based dosage optimization should be considered.
R: We have added a Table in the manuscript including the most important studies describing differences in therapeutic treatment response among sexes. To our knowledge, do not exist studies performed by choosing the drug dosage according to sex. However, we have included in the table the overall dosage for each described drug.